# ModulOM: Disseminating Deep Learning Research with Modular Output Mathematics

**Maxime Istasse,**\* **Kim Mens, Christophe De Vleeschouwer**
ICTEAM institute, Université catholique de Louvain, Belgium
`{maxime.istasse,kim.mens,christophe.devleeschouwer}@uclouvain.be`

## Abstract

Solving a task with a deep neural network requires an appropriate formulation of the underlying inference problem. A formulation defines the type of variables output by the network, but also the set of variables and functions, denoted output mathematics, needed to turn those outputs into task-relevant predictions. Despite the fact that the task performance may largely depend on the formulation, most deep learning experiment repositories do not offer a convenient solution to explore formulation variants in a flexible and incremental manner. Software components for neural network creation, parameter optimization or data augmentation, in contrast, offer some degree of modularity that has proved to facilitate the transfer of know-how associated to model development. But this is not the case for output mathematics. Our paper proposes to address this limitation by embedding the output mathematics in a modular component as well, by building on multiple inheritance principles in object-oriented programming. The flexibility offered by the proposed component and its added value in terms of knowledge dissemination are demonstrated in the context of the Panoptic-Deeplab method, a representative computer vision use case.

## 1 Introduction

Proper dissemination of deep learning research outcomes requires pedagogical explanation of theoretical concepts, but also transfer of hands-on experiments (Isensee et al., 2021). Papers succeed in the former, but are limited regarding the second. Despite Github repositories and popular deep learning libraries (e.g. PyTorch (Paszke et al., 2019), Tensorflow(Abadi et al., 2015)) nowadays provide a common way to share code and information regarding the practical implementation of a model, the release of code is still far from solving all issues related to reproducibility, fair comparison of methods and, maybe more importantly, transmission of knowledge required for fruitful incremental development. This is because the multiple code bases released to address a given task often vary in the way they define and manipulate the variables associated to their respective mathematical formulation of the task, i.e. in the variables and functions, denoted output mathematics, defined to turn the network outputs into task-relevant predictions (e.g. compare the predictions in (Cheng et al., 2020) and (Neven et al., 2019)). This makes their comparison subtle since involved data, and intermediate results (of interest for debugging and/or understanding purposes), may largely differ. It is frequent that the relatively minor variations of the task formulation reported in a paper lead to significant changes in the associated reference software implementations.

Fundamentally, those large discrepancies between implementations are primarily due to the lack of modularity associated to the processing of the network outputs. Modularity is a key desired feature when designing software. Decomposing a program in modules improves understandability, makes the code easy to reuse, facilitates its evolution and/or maintenance, and allows multiple programmers to work independently on distinct modules (Parnas, 1972).

Without surprise, modularity is ubiquitous in the code base of any deep learning (DL) experiment. For instance, external libraries often provide different optimizers, datasets, data augmentation def-

---

\*Code is available at `https://github.com/mistasse/modulom-panopticdeeplab`

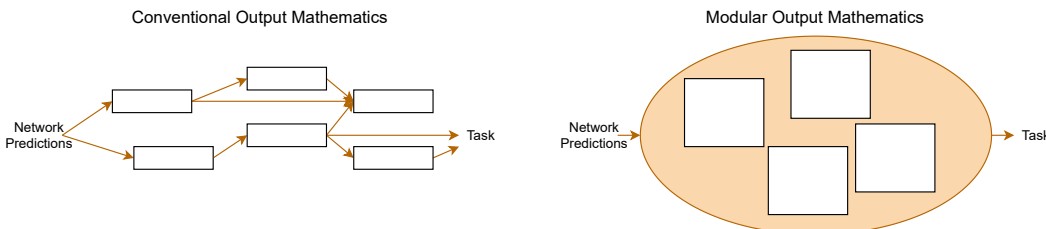

Figure 1: Implementation of output mathematics (OM) : (left) Traditionally, OM are implemented as a set of functions whose calls have to be carefully orchestrated by external code because of the flow of data enforced by their signatures, as depicted by orange arrows. (right) Our solution consists in splitting the OM into classes in object-oriented programming. Those are assembled using multiple inheritance to create the OM component. Variables of interest to the OM are accessed through class methods, so that they become available anywhere in the OM, without requiring intricated signatures and dependencies. This is depicted by the orange pool, in which modules can freely access any data.

initions that can be used off the shelf to create interchangeable components (Chollet et al., 2015; Paszke et al., 2019). This is also the case for the definition of neural networks architecture. In that case, residual blocks, convolution or pooling layers can easily be combined and substituted to define network variants (Chollet et al., 2015; Paszke et al., 2019), supporting the emergence of automatic search for neural architectures (Jin et al., 2019).

In contrast, the mathematics related to the network output are generally not implemented in a modular manner. Those mathematics include the functions that define losses, the functions required to translate the network output into semantically meaningful information, and potentially many other functions that compute intermediate variables or are required to visualize the manipulated concepts (e.g. accuracy or quality metrics, but also high-dimension tensors visualized based on a PCA transform), for debugging purposes for instance. In the rest of this manuscript, the term *output mathematics* is used to denote this set of mathematical functions that handle the network outputs.

The reason why output mathematics (OM) are not modular lies in the fact that they are conventionally implemented as a set of functional abstractions (i.e. functions). Fundamentally, functional abstractions are constrained by their signatures, i.e. their list of arguments, and are thus especially suited to deal with scenarios where the developer wants to test multiple versions of a single function whose interface with the rest of the program is fixed. This scenario does not fit to the OM case, often characterized by the manipulation of multiple interdependent functions, especially in modern computer vision models (pose estimation (Newell et al., 2017; Papandreou et al., 2018; Kreiss et al., 2019), instance segmentation (Neven et al., 2019; Novotny et al., 2018; He et al., 2017) or panoptic segmentation (Kirillov et al., 2019; Cheng et al., 2020)). This interdependency requires careful orchestration to feed each function with the right arguments. Hence, as depicted in Figure 1 (left), the functions written in the code base of such experiments end up in being largely constrained by their signatures, which determine the way they access the variables they process. They depend on each other and are not plug-and-play: the alternatives to a function are limited to those with the same interface, which can be designed to be called at the same time with the same dependencies. Consequently, the changes in dependencies induced by slight modifications in the task formulation might impact the code in many places. This hampers the deployment of a modular implementation paradigm, which penalizes transmission of know-how and incremental research, since it prevents leveraging existing OM methods to investigate alternative task formulations. Promoting modularity would make sub-tasks free to evolve in unforeseen ways at minimal code rewriting cost, but would also help in comparing alternative task formulations by pointing explicitly the modules that differ between them.

This paper proposes a first practical solution to make the OM modular, and illustrates how the underlying structure of such a modular framework favors fluent dissemination of research outcomes, enabling easier increments and more accurate comparisons.

Our prime contribution lies in a set of recommendations to facilitate developers' work when exploring the space of formulations to address a given task. As illustrated in Figure 1 (right), the key idea of our proposal consists in making the OM variables accessible to methods without constraining

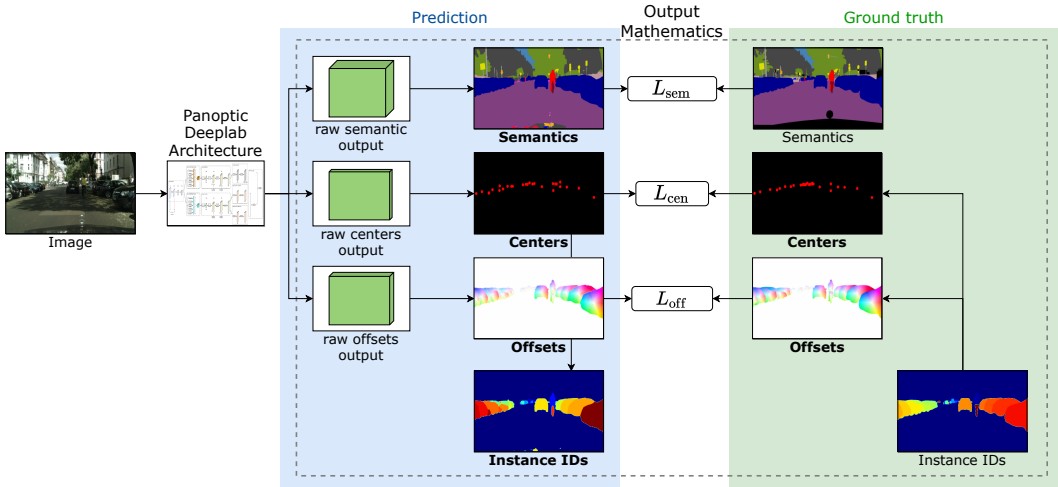

Figure 2: Diagram of the Panoptic-Deeplab method illustrating its mathematical variables and their flow. Some of those mathematical variables are provided by the dataset (*e.g.* image, ground-truth instance IDs), some are provided by the output of the neural network (*e.g.* raw semantic, centers and offsets outputs), while the other variables, indicated in bold, have to be computed from those for training the model or interpreting its results. We propose to regroup all the variables inside classes so as to make them easy to access, interchange and modify. (see Figure 7)

function signature. This is achieved by defining the OM modules as classes, and by making all OM variables accessible to the class method through dedicated methods. Multiple class inheritance principles, with method overriding and mixins, are shown to offer a natural way to increment the code or explore variants of the problem formulation. Our proposal, grounded in the class inheritance paradigm, is a step forward in offering modularity, with minimal constraints on functions interfaces and inter-dependencies. By relaxing the assumptions made about how those functions interact with the rest of the code (both in terms of access to data and function composition), it facilitates the presentation of the output mathematics as an inheritance diagram that explicitly presents how methods differ from a baseline, with the purpose of facilitating both the transfer of knowledge and the incremental development of DL models.

In a nutshell, through appropriate definition of classes and hierarchical dependencies, multiple inheritance is shown to offer the flexibility needed to overwrite and/or compose functions at will, making the output mathematics module easy to adapt or substitute by another.

The rest of the paper is organized as follows. Section 2 surveys the inheritance mechanisms that are mobilized to implement our solution. It also concretely defines the notion of output mathematics. Section 3 presents our proposed paradigm to implement the output mathematics as a modular component, and explains why and how this paradigm overcomes most of the issues encountered by the conventional and scattered implementation alternatives. Section 4 discusses key implications of modular output mathematics for research dissemination. Section 5 concludes.

## 2 BACKGROUND

### 2.1 NETWORK OUTPUT MATHEMATICS

Fundamentally, in a deep learning model, the output mathematics correspond to the set of operations that are related to the transformation of the output activations of a neural network into semantically meaningful information. They are defined in mathematical language, through the manipulation of mathematical variables. They include (i) the definition of the training loss(es), typically comparing the network output(s) with some representation of the desired prediction(s), but also (ii) the functions that process/decode the (multiple) outputs of the network to turn them into meaningful predictions.

For example, modern pose estimation methods, e.g. (Kreiss et al., 2019), predict a set of heatmaps in which hot-spots locate a specific type of keypoint (e.g. head or feet), together with vector fields that are used to associate the keypoints of an instance. As another example, popular instance segmentation methods predict a seed map to locate the instance center, together with a map of embeddings, defined so that the pixels of individual instances aggregate in tight and distinct clusters in the embedding space (Newell et al., 2017; Neven et al., 2019). In this formulation, each instance center corresponds to a hot-spot in the seed map, and the pixels of a specific instance are simply identified as the ones whose embedding lies sufficiently close to the embedding of that particular instance centers. Figure 2 illustrates this principle in the case where the embedding space corresponds to 2D spatial coordinates, defined in each pixel as the sum between the pixel coordinates and a 2D offset that is predicted by the network. In this scenario, the network is trained to make the embeddings of all pixels associated to a same instance equal to a common 2D vector, referred to as the instance center. This corresponds to the Panoptic-Deeplab method, defined in (Cheng et al., 2020). The figure presents the main variables involved in the instance segmentation process. The variables depicted in blue on the left-side are directly predicted by the network, in the form of raw tensors. Other variables are computed from the dataset (ground-truth column), from the raw network predictions (predicted maps column), or from both the dataset and predicted maps (loss functions). The mathematical operations that are used to compute all those variables (i.e. maps and losses) correspond to the so-called output mathematics, and directly reflect the computer vision problem formulation. In Section 3, we present a number of implementation guidelines that offer the flexibility required to change those functions at will. The application of those guidelines to implement and compare variants of the Panoptic-Deeplab formulation is presented in Appendix to demonstrate the advantages of our proposal in terms of incremental development and knowledge transfer. Since those guidelines build on the multiple inheritance paradigm, we briefly review its key principles in the next section.

## 2.2 MULTIPLE INHERITANCE

Fundamentally, in object-oriented programming, classes are extensible templates for creating objects, providing the initial values for object instance variables, and the bodies for methods (Bruce, 2002). All objects generated from the same class share the same methods, but manipulate distinct instance variables. New objects are created from a class by calling its constructor with specific arguments.

A class can inherit the base behavior of another class. Its methods are defined by default to those of its base class, but they can also be overridden to refine their behavior as needed. For compatibility reasons, the overriding method usually has the same name, parameters, and return type as the method it overrides. Overriding helps in avoiding code duplication between different classes that would only differ through subtle variants of their methods.

For this reason, classes are a construct of choice when engineering modular systems. A base class can define the generic behavior of a component as well as how it should interact with the external world, while leaving some methods to be refined by its child classes, created for a specific purpose. In the early age of object-oriented programming, it was not well understood how a class should behave if it had to inherit from several classes implementing a same method differently. For this reason, and because it proved to be sufficient to design most systems, many languages only allowed single inheritance, and engineers learned to design systems that way.

However, as illustrated in Figure 3, multiple inheritance largely increases object-oriented programming flexibility. It enables an object or class (here, D and E) to inherit characteristics and features from more than one parent (here B and C). To address the ambiguity as to which parent a particular method (here $f$) is inherited from when more than one parent class implements the said method, composition mechanisms have been proposed. In Python, the order of inheritance affects the class semantics, and the ambiguity is solved using the C3 linearization (or Method Resolution Order (MRO)) algorithm (Barrett et al., 1996). This gives the language the opportunity to support multiple inheritance with consistent and well-understood behavior.

This solution to disambiguate multiple inheritance makes it possible to exploit *mixins* (Bracha & Cook, 1990), which denote classes (e.g. B and C in Figure 3) that are defined with the purpose of being combined with a base class. The combination of a mixin with a set of base classes results in a family of modified classes (like D and E in Figure 3). It allows a programmer to share a functionality

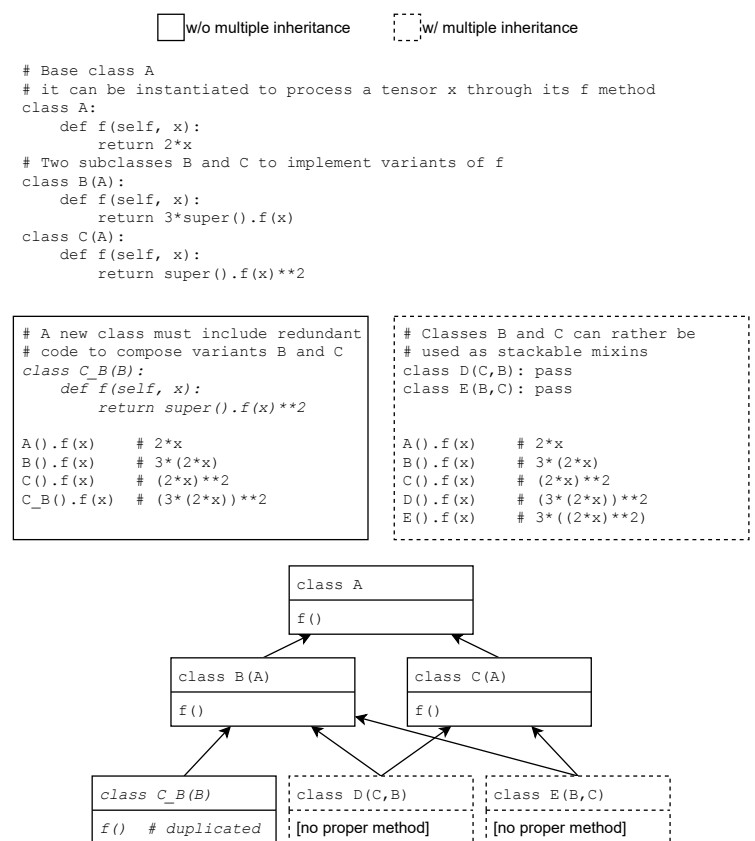

Figure 3: An example of the flexibility brought by multiple inheritance. Single inheritance does not make it trivial to create a module with B and C features at the same time. One solution could imply to duplicate the code of one in a child class of the other (as done here by C_B), but this raises maintenance issues. In contrast, multiple inheritance enables the mixing of both features in a new class without requiring additional code, nor structurally complex code.

between multiple classes without duplicating the same code in them. An example is illustrated in Figure 3 where a single mixin (B or C) is created to affect some methods (here *f*) in a precise way without having to know exactly into which base class it will be combined.

When a class inherits from several mixins, the modifications induced by the mixins are composed according to the order in which the mixins are listed in the inheritance declaration. See Figure 3 for examples.

## 3 MODULAR IMPLEMENTATION OF OUTPUT MATHEMATICS

Our paper proposes a solution to manipulate the network output mathematics as a modular component. Our solution aims at promoting reuse and incremental refinement, across experiments, of the functions involved in the mathematical formulation of the task handled by the network. It is expected to support incremental and experimental development, but also to facilitate the exploration of various formulations of the optimization problem addressed by the training phase, e.g. based on different losses. Those alternative formulations (e.g. of the computer vision problem of interest) might even consider the prediction of alternative variables than the ones considered in the baseline reference formulation.

Our proposal should be considered as a reference baseline, to be challenged and refined in future research. It aims at enabling fluent composition and replacement of OM methods, making it possible to share hands-on experiments, and to compare between distinct formulations of the relation

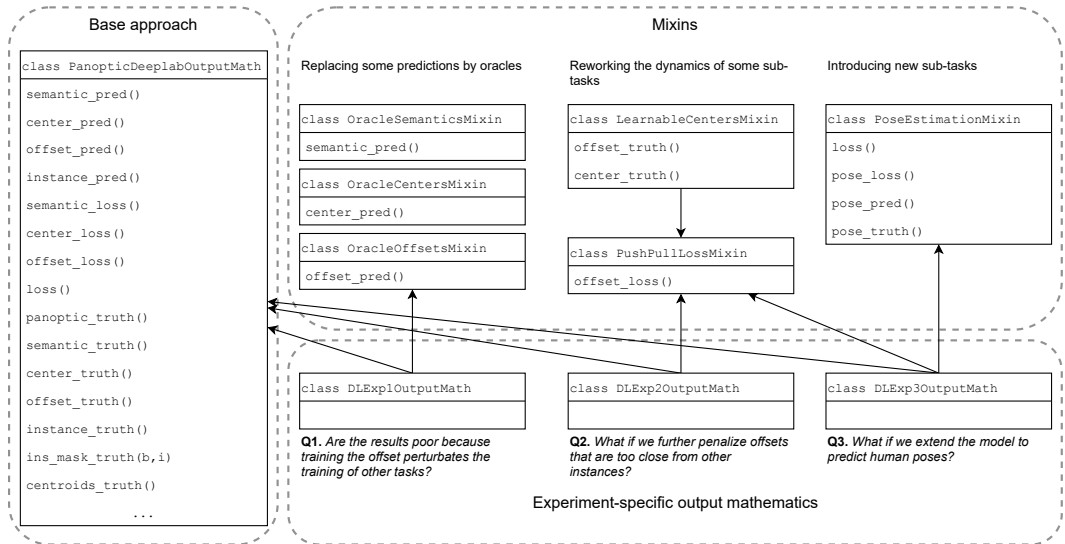

Figure 4: A use-case example in which our two recommendations are applied to make the output mathematics flexible and its code reusable across many formulations. The same code, augmented with a few mixins, is used to investigate 3 questions (Q1, Q2, and Q3) related to the behavior or the extension of a baseline, without having to perform any modification to the code bases of experiments (except maybe for additional heads of the neural architecture in the case of an additional sub-task, which are most often modular anyhow).

between the network prediction and the task at hand. In practice, our solution consists of a number of implementation guidelines that can be summarized as follows.

Our first recommendation consists in assigning a specific class to each DL experiment, i.e. to each problem formulation, hosting as methods the mathematical functions implied in the computation of the task results and associated losses : the experiment's output mathematics class. To minimize code duplication, this class should inherit from mixins the behaviors that deviate from a baseline formulation. The baseline should thus be defined to implement methods shared across multiple modified experiments. This is illustrated in Figure 4 for the use case introduced in Figure 2, and further studied in Appendix B.

The second recommendation addresses the data accessibility. To reduce the constraints and inter-dependencies induced by the inclusion of arguments in the signature of methods, the variables that are relevant to the output mathematics, including ground-truth, raw predictions, and high-level vari-ables derived from the latter, should be part of the object instantiated from the output mathematics class and be made accessible through its methods with as few parameters as possible. In that way, those variables become available to all the other methods of the object, without having to anticipate their usage within a list of arguments in the signature of each method. This offers an increased flexibility when reusing or overriding a method.

This flexibility is illustrated in Figure 5, which depicts an example where a method is overridden by the child class. On the left side, each method receives a particular variable as parameter, which causes an issue when the child class method wants to use another variable than the one considered in the base class. In contrast, on the right side, all variables that are relevant to the studied problem are part of the class' instance variables and are available through dedicated methods, without the need to be passed as arguments. As a consequence, the child class method can be adapted whatever the variable required by its new behavior. In more detail, the goal here is to create a mixin replacing the predicted semantic segmentation by the ground-truth semantic segmentation. Such replacements can be useful for ablation studies for instance, to reevaluate a metric in case a sub-task is perfectly solved, or to evaluate whether the training of a particular sub-task perturbates that of others. In this situation, we can observe on the left side that the ground-truth segmentation cannot be returned by the `OracleSegmentationMixin` simply because the ground truth has not been routed as an argument of the inherited `semantic_pred` method. Indeed, based on the base class, external code

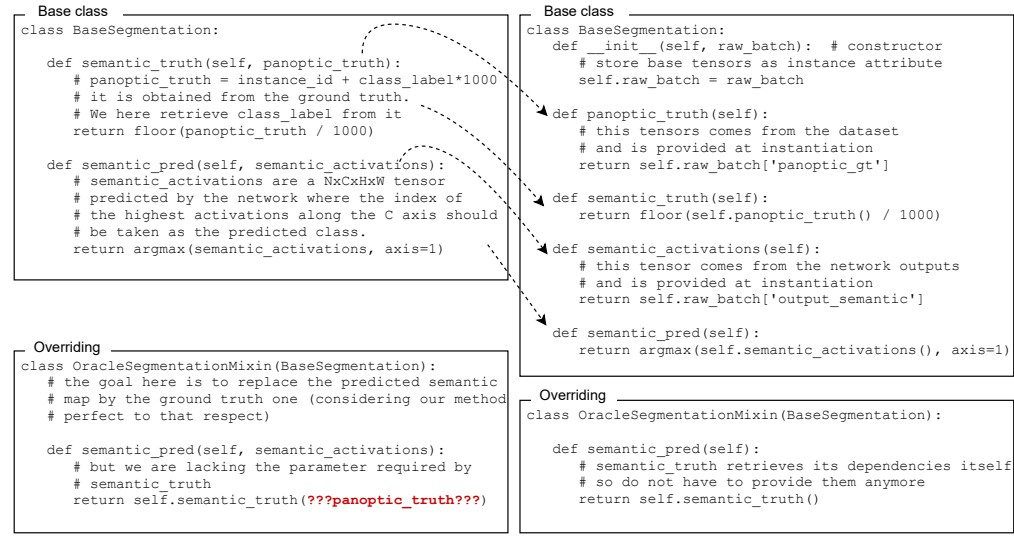

Figure 5: (left) Situation where the traditional parameter passing of variables prevents a mixin to implement a desired feature. The goal here is to create a mixin replacing the predicted semantic segmentation by the ground-truth semantic segmentation. The semantic_pred and semantic_truth methods respectively receive the semantic_activations and panoptic_truth parameter. Therefore, mixins cannot expect their methods to receive other parameters. It makes it impossible for OracleSegmentationMixin to redefine semantic_pred so that it returns the semantic_truth variable. (right) By making variables accessible through methods calls, passing variables as function arguments is not necessary anymore. This reduces the constraints imposed by the method signatures and renders the above-described modification trivial.

is only providing to the semantic_pred method the semantic_activations parameter, not the panoptic_truth required by the semantic_truth. It is not rare to encounter similar situations in less trivial examples: predicting uncertainties, integrating new tasks in the losses, or defining a variable as a function of the network output rather than as a constant derived from the dataset ground truth. This last case occurs for example when replacing the constant ground-truth instance center by the mean of the embeddings associated to the ground-truth instance (Newell et al., 2017; Neven et al., 2019).

To avoid duplication of computation, we add a memoization mechanism that caches the result on the first call and restitutes it on subsequent calls. Systematic caching means that any variable computed on a batch stays in memory for as long as that batch. Though this may sound naive, it often makes sense in the context of deep model training since most intermediate results (among which neural network activations) have to be remembered for the backpropagation anyway. Those activations alone usually outweigh by a large margin most of the byproducts required to compute the loss. Furthermore, not caching a variable in the output mathematics may result in computing it many times, which potentially also leads the backpropagation library to store it multiple times, thereby increasing the memory-footprint.

It can however make sense to allow the programmer to introduce exceptions to this rule. Such a mechanism has been implemented in the example code provided with this paper. (Istasse, 2021)

Appendix B considers a representative example of a computer vision deep learning algorithm to challenge our assumptions about the limited memory and computational impact induced by our proposed implementation paradigm on training performance. Beyond showing that the overhead is far from prohibitive, the work reported in this Appendix first reveals that our proposed recommendations for modular output mathematics can be naturally implemented in the reference Panoptic-Deeplab method, with minimal changes to the reference code. Second, the modular output mathematics implementation is leveraged to prototype a variation of the instance segmentation problem formulation, incorporating learnable instance centers in the instance prediction process. This work reveals that,

thanks to its modularity, the output mathematics component is straightforward to adapt to the new formulation of the problem.

## 4  DISSEMINATION OF OUTPUT MATHEMATICS

Figure 7 illustrates how the modular organization of the output mathematics proposed in this paper helps in disseminating deep learning experiments, and favors incremental development and fair comparisons of methods. This figure provides an example of how to document the reference software released with a deep learning experiment, by defining explicitly the link between the mathematical formulas introduced in the paper associated to the DL experiment, and the variables and methods manipulated by the reference software. It presents how the variables and the mathematical formulas are (i) grouped into classes and (ii) defined as class methods in the code. In this list of correspondences, other researchers will find the information required to access and modify particular variables and methods. Upon incremental development, researchers just have to prepare and disseminate a table gathering the classes and methods that have been modified compared to the baseline. This is exemplified in Figure 7 for the Panoptic-Deeplab method with learnable centers.

Note that since variables are accessed without resorting to constraining function signatures, data are arbitrarily available, making it easy to create unforeseen alternative formulations of the methods, without the need to describe complex interdependencies related to the data flow. Furthermore, modularity enables creating banks of plug-and-play variants for a given baseline, in a way analogous to what is possible with neural architectures, optimizers, or data augmentation. Finally, it also becomes possible to extract the output mathematics of a baseline (and therefore the complete set of variants built on it) from one project and inject them in another as only some key inputs and outputs (mostly the task variables in Figure 7) need to be interfaced with external code.

## 5  CONCLUSION

This paper has proposed to group the functions implementing the output mathematics (OM) inside a modular component, offering increased flexibility when designing deep learning experiments, by allowing easy substitution and modification of the mathematical functions associated to the problem formulation handled by the neural network.

Our paper focuses on the OM because this part of the code is characterized by a potentially large number of interdependent functions, dealing both with network- and task-related data. This is in contrast to most other DL components are related either to the network (e.g. optimizer, scheduler, or network layers) or to the task (e.g. data augmentation or task-related metrics and visualization routines), and are characterized by well-defined interfaces that make their functional abstraction relatively straightforward.

Our modular implementation of the output mathematics builds on two main recommendations that can be summarized as follows. First, by implementing the output mathematics as methods associated to a set of base and mixin classes, various problem formulations can be designed in a modular way by leveraging the multiple inheritance paradigm. Second, by exposing the mathematical variables involved in the problem formulation through methods with only necessary parameters, those variables become available to all other methods, relaxing the constraints and dependencies imposed by method signatures over the code and the data flow.

Application of those guidelines to the Panoptic-Deeplab method has revealed that our proposed modular implementation is easy to integrate in an existing experimental code base, without penalizing its performance. The flexibility of the proposed approach has also been demonstrated by showing how easy it is to change the problem formulation, the definition of associated variables and the data flow.

We believe this work provides a convenient way for practitioners to run the many different experiments involved in the development of DL models. It may also facilitate the comparison of similar formulations. Moreover, because it gives access to new formulations from a limited amount of code, it opens perspectives regarding the automatic exploration and optimization of the output mathematics, as is done for some other parts of the DL experiments such as the data-augmentation pipeline and the neural architecture (Cubuk et al., 2019; Jin et al., 2019).

ACCESSIBILITY STATEMENT

The ideas presented in this paper aim at more accessible research, by making it easier to build on an existing model and to compare different methods on the same ground. The authors hope that the guidelines proposed to write and disseminate the code related to deep learning experiments will lead to widespread solutions, e.g. that they will get adapted to other languages, or that mainstream libraries will adopt similar constructs. The authors will be glad to collaborate to make it happen.

The authors are willing to make the content of this publication accessible to the widest possible audience who could benefit from it. Our example source code (Istasse, 2021) is released along with a comprehensive *readme*. For help requests or suggestions, please do not hesitate to open an issue on the GitHub repository or to contact the authors by e-mail.

ACKNOWLEDGEMENTS

This research is supported by the DeepSport project of the Walloon Region, Belgium. C. De Vleeschouwer is a Research Director of the Fonds de la Recherche Scientifique - FNRS.

Computational resources have been provided by the supercomputing facilities of the Université catholique de Louvain (CISM/UCL) and the Consortium des Équipements de Calcul Intensif en Fédération Wallonie Bruxelles (CÉCI) funded by the Fond de la Recherche Scientifique de Belgique (F.R.S.-FNRS) under convention 2.5020.11 and by the Walloon Region.

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

## A  DATAFLOW GRAPH CONSTRUCTION

By including the variables of interest in the OM class, the rest of the experiment code base does not have to care about their routing. The output mathematics component becomes self-sufficient for transforming provided inputs into all the products that can be derived from them. All those transformations and associated routing of data are equivalent to a dataflow graph which is encoded in a class, defined from a base class and mixins. To apply those transformations to actual inputs, the class is instantiated based on the tensors associated to those inputs (and forming the *raw batch*), and the value of a variable of interest can be obtained by calling the instance method associated with it. Figure 6 illustrates the dataflow graph that is inferred from the code on the right side of Figure 5.

Without the modular output mathematics component, other components of the experiment that are external to the OM have to call the OM functions with variables to be passed as arguments. Therefore, variables are routed by external code that could transform them, which prevents the inference of a similar dataflow graph from the OM code alone.

Moreover, because of the careful routing that has to be maintained when variables are passed as arguments, it is costly to ensure that a same variable is never computed twice at different places. In

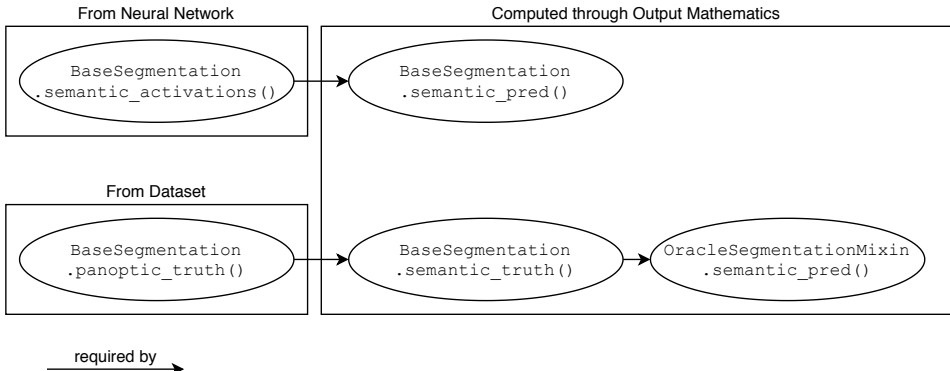

Figure 6: Dataflow graph inferred from the right side of Figure 5. Each node is a variable that can be obtained from the output mathematics module through a method call. Upon call, the body of the target method will be executed, and will itself retrieve the data required for the computation of its result (any node having an edge pointing to it). This graph is fully determined by the module, which is not the case when the functions receive data as arguments, because those data are subject to computations by external code.

contrast, handling variables through method calls makes it easy to force any variable to be computed at most once. Each method can indeed be adapted in order to cache its result on a first call in an instance attribute, and to retrieve the cached result on subsequent calls with identical parameters. In case a variable is not required, because the variable is not used in the current variant or mode (*e.g.* training or prediction), the method is simply never called, and resources are not wasted to compute it.

One improvement over the recommended systematic caching policy would be to release each variable when it is used for the last time, or to recompute terms that are not required by backpropagation. Those conditions are currently difficult to detect given the interpreted nature of Python, but when the whole code having access to the output mathematics component is known, those problems can be addressed through dataflow analysis and optimization. [1]

In addition to those concerns related to potential memory overhead, caching mechanisms and calls through an entire hierarchy of classes introduce extra instructions to be performed by the CPU. However, in practical cases, the mathematical computations required by the deep learning algorithm are much more expensive than those mechanisms, so that the overall time overhead of the latter is shown to be negligible[2] (see experiments in Appendix B).

## B  ASSESSMENT ON THE PANOPTIC-DEEPLAB METHOD

Panoptic-Deeplab is a pixel-wise panoptic segmentation approach that, although surprisingly simple, achieves state of the art performance.

This use case has been chosen because it provides a popular and representative deep learning solution to a fundamental computer vision problem. The integration of our modular component to this case will be released in the form of a complete repository in (Istasse, 2021), along with the camera-ready version. It first reveals that an arbitrary code can conveniently be adapted to host our proposed

---

[1]Despite this question lies outside the scope of this paper, the easiest way to benefit from such optimizations is to use our module to create a static computational graph, to be evaluated later on every batch, as can be done using Tensorflow (Abadi et al., 2015). In such a case, practitioners can use our component to initialize the computational graph, instantiating it with placeholder variables in the raw batch and creating variables in the graph by calling their associated methods in the component. Such a graph is a low level structure where every single instruction required to compute a variable is known in advance. At execution, it is provided a set of input tensors and the set of requested variables, so that its library can compile only that part of the graph and optimize the data flow accordingly.

[2]Here also, this overhead would only apply at initialization if our component was used to create a static computational graph.

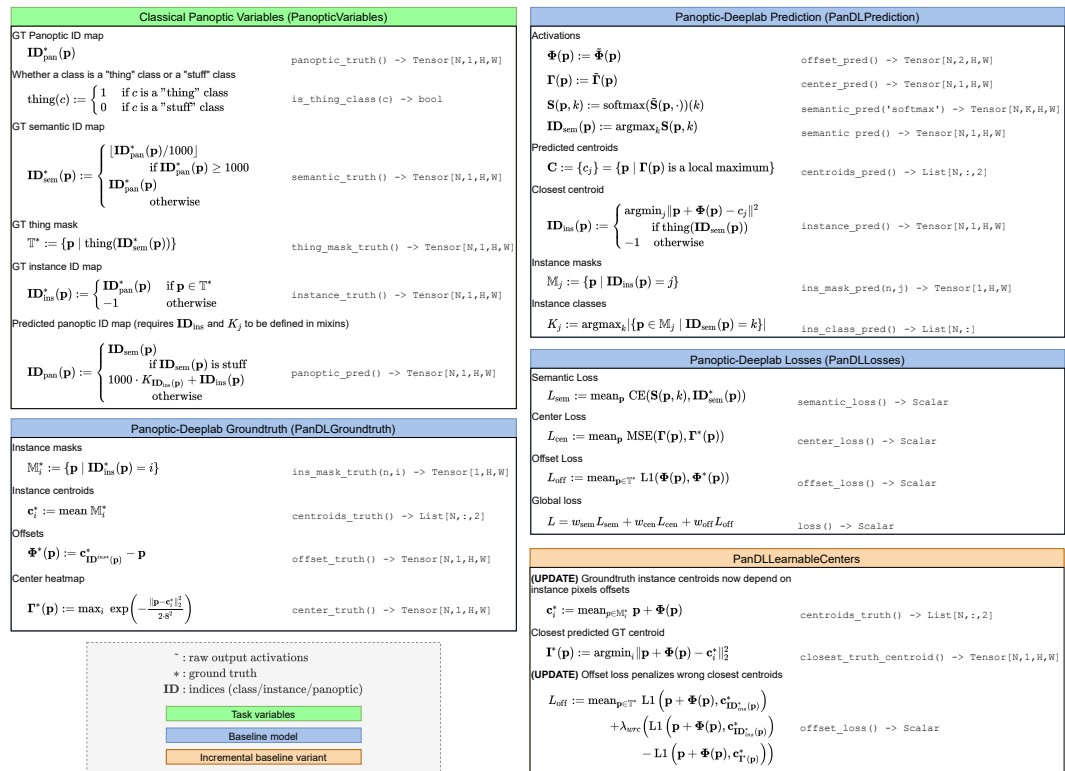

Figure 7: Illustration of an implementation of the baseline Panoptic-Deeplab output mathematics and of its incremental extension (to support learnable centers), following our recommendation to exploit multiple inheritance for modularity. The different blocks, respectively devoted to variables and OM functions definitions, are classes that can be inherited by the final output mathematics of an experiment. Those blocks consist in a set of lines that each associate a variable introduced to formulate the task in the paper to its corresponding method in the code. The green block includes task specific variables, i.e. variables that are relevant whatever the mathematical formulation. Blue blocks present the variables manipulated by the baseline reference method, while the orange block introduces the novel variables introduced to increment the baseline with learnable centers.

modular and composable implementation of the output mathematics, with limited resource overhead. It also illustrates how the flexibility offered by our solution facilitates the communication about the incremental development of an alternative formulation of the instance segmentation problem.

To demonstrate the feasibility and the advantages of our proposal, we proceed in two steps. First, we apply our recommendations for modular output mathematics on the public Panoptic-Deeplab reimplementation. We observe that this can be done easily, with minimal changes to the reference code. Second, we exploit our modular output mathematics to prototype a variation of the instance segmentation formulation, incorporating learnable instance centers in the instance prediction process. This work reveals that, thanks to its modularity, the output mathematics component is straightforward to adapt to the new formulation of the problem. Figure 7 illustrates how we have chosen to formulate the Panoptic-Deeplab method using modular output mathematics and to extend them for the learnable centers.

Time and memory overheads, compared to the base implementation, are estimated for experiments running our modular output mathematics component, with and without the added learnable centers functionality.

## B.1 Modular output mathematics

Integrating our recommendations for a modular implementation of the output mathematics into the public Panoptic-Deeplab implementation (Cheng et al., 2020) is fairly straightforward. Details about

our specific implementation choices can be found in the code (Istasse, 2021). Here, we survey the main steps involved during this development.

The first step consists in implementing the four upper classes illustrated in Figure 7, associating a class method to every mathematical concept present in the initial code. Sometimes, a same concept can be represented in different ways and it is difficult to come up with different names. In such cases, it can be practical to handle the different representations of a concept in a method with a "representation" parameter. For instance, the semantic maps are predicted by the network as unbounded activations maps, from which probabilities can be obtained via the application of the softmax function, and the predicted class through the argmax function. We use the representation parameter to refer to them as `semantic_pred('logits')`, `semantic_pred('softmax')` and `semantic_pred('argmax')` respectively.

The output mathematics class is in our case an anonymous class created once and for all at the beginning of the execution, inheriting from `PanDLBase`, itself inheriting from the four aforementioned classes, and the eventual mixins specified by the practitioner in the experiment configuration. Each raw batch is used as a parameter to the constructor in charge of instantiating the class to create an instance specific to that set of data.

Each obtained instance can then be forwarded to the parts of the code requiring access to tensors, *e.g.* to perform the optimization step, to compute and log metrics, or to draw visualizations. In our case, we reused the argument dedicated to the raw batch. Finally, data computations in those parts of the code only need to be replaced by calls to the adequate methods of our output mathematics instance. This way, those components that refer to the output variables do not contain code that is specific to Panoptic-Deeplab anymore. Hence, they become compatible with alternative implementations of the output mathematics.

To handle the caching, we make the `PanDLBase` class also inherits from the `OutputMath` class that implements the default memoization policy described in Section 3 while providing ways to specify exceptions to it.

## B.2 Implementing the LearnableCenters mixin

Formulating the instance segmentation problem based on learnable instance centers implies that the network is not explicitly supervised to predict centroids at the center of instances anymore. Instead, each ground-truth instance center location depends on the output of the network, and get positioned at the center of mass of the locations $\mathbf{p} + \mathbf{\Phi}(\mathbf{p})$ pointed pixels of that instance. This way, the network is given the freedom to redistribute instance centers spatially, which may help to solve clustering ambiguities when some instances have centers that are close to each other. This idea is common in the literature (Newell et al., 2017; Neven et al., 2019).

Were the ground-truth offsets and center heatmap computations implemented conventionally, adjusting them would have required heavy rearranging of the code and its data flow. In the public Panoptic-Deeplab reimplementation for instance, those targets are computed on the CPU before sending data to the GPU and running the neural network, and therefore have no access to the latter's predictions.

In our case, it appears, as suggested in Figure 7, that only overriding the method associated with the $\mathbf{c}_i^*$ variable is required. For better centroid separation however, we also choose to adapt the offset loss with a term that penalizes pixels pointing to locations $\mathbf{p} + \mathbf{\Phi}(\mathbf{p})$ lending closer to other centroids than that of their ground-truth instance.

In the next section, we assess that the overhead introduced by our modular output mathematics is not prohibitive with respect to their benefits.

## B.3 Overhead induced by the modular framework

Four experiments have been run and compared: (i) the original Panoptic-Deeplab training loop, where the target maps (i.e. ground-truth centers, offsets and semantic labels) are computed on the CPU; (ii) an equivalent training, with our proposed modular output mathematics but accessing the targets maps that are still computed on the CPU; (iii) a training with target maps being computed in

Table 1: Average ($\pm$ standard deviation) duration of the forward pass, the backward pass, overall batch and the peak GPU memory usage over 100 batches. Four scenarios were considered: the original method with the original code, the original method with a shallow integration of our OM component (the targets are still computed on the CPU), the same with target computation integrated in the modular OM (on the GPU), and a variant of the method with learnable instance centers.

| Experiment | Forward pass (s) | Backward pass (s) | Overall batch time (s) | Peak memory (GB) |
|---|---|---|---|---|
| Original Panoptic-Deeplab | $0.290 \pm 0.004$ | $0.471 \pm 0.019$ | $5.370 \pm 1.566$ | $26.230 \pm 0.000$ |
| Panoptic-Deeplab w/ ModulOM | $0.293 \pm 0.009$ | $0.474 \pm 0.053$ | $5.390 \pm 1.476$ | $26.246 \pm 0.001$ |
| Panoptic-Deeplab w/ ModulOM w/ targets in OM | $0.387 \pm 0.029$ | $0.477 \pm 0.008$ | $1.010 \pm 0.099$ | $26.640 \pm 0.043$ |
| Panoptic-Deeplab w/ ModulOM w/ targets in OM w/ learnable centers | $0.667 \pm 0.655$ | $0.556 \pm 0.030$ | $1.500 \pm 0.819$ | $27.427 \pm 0.301$ |

the OM component, on the GPU; and (iv) a training with the learnable centers mixin replacing the fixed centers in the OM.

In each of those, we track the duration of forward pass, backward pass, both happening on the GPU, and the overall time of a training batch, which includes computing the targets in CPU threads when needed. We are also interested in the maximal GPU memory allocated, since it generally constrains the batch size.

We perform those experiments on a machine with one NVidia V100 GPU, a batch size of 4, and dedicate 4 CPU cores to data loading (of an Intel Xeon Gold 5217). All the other parameters come from the MobileNetV2 configuration of Panoptic-Deeplab. The results are averaged over 100 consecutive batches.

Table 1 reports each metric for all experiments. The only difference between (i) and (ii) is the integration of the OM component. A significant amount of computations still happen outside the OM because the target maps are computed by the data-loader CPU threads. Both rows exhibit very similar timings and peak memory usage, which confirms that the overhead introduced by the component, its calls and caching mechanisms, is negligible. Implementation (iii) moves the computation of targets from data-loading happening in CPU threads to the OM component. The overall batch processing time is heavily reduced compared to (ii), because less data is required to be transferred between CPU and GPU, and because the GPU is a more efficient device to perform those operations, despite the many loops that have to take place. Would it not be the case, it would still be possible for the programmer to specify that some computations inside the OM have to take place on the CPU. We emphasize that, as suggested in Section 3, despite the systematic caching of intermediate results taking place, the increase in peak memory is relatively low. Finally, (iv) adds the learnable centers on top of (iii). The overhead of this particular variant is significant because it fundamentally impacts the mathematical system, using network predictions for creating targets, hence creating new operations contributing to the gradients. In addition, the computation of the new embedding loss sometimes has to occur on the CPU because computing the distance between the embedding of each pixel and each ground-truth instance centroid requires more memory than available on the GPU. Yet, all-in-all, the modularity of our output mathematics allows us to implement it in a way that it is still on average more than three times faster than the public Panoptic-Deeplab implementation.

We see the fact that this complexity could be added seamlessly, contained in a class definition, as an example that the proposed modular OM are effective to broaden the possibilities of existing code bases in a practical fashion.

