# OpenReview forum: "ModulOM: Disseminating Deep Learning Research with Modular Output Mathematics"
_ICLR.cc/2021/Workshop/Rethinking_ML_Papers — Rethinking ML Papers - ICLR 2021 workshop Poster_

### Official Review · AnonReviewer1 · 2021-03-28
**Review for "ModulOM: Disseminating deep learning research with modular output mathematics"**

**Accessibility:**

Score of 4 (Strong): Submission states accessibility concerns and provides solutions within the proposed framework. However, it does not declare the limitations and exceptions.

**Litreview:**

Score of 4 (Strong): The submission directly differentiates itself from previous works and formats.

**Problemstatement:**

Score of 4 (Strong): The submission sets a very strong example of how to address the problem, which should be relevant to the workshop themes.

**Relevance:**

Score of 3 (Neutral): Attempt was clearly made to address a theme of the workshop, but it seems that the work was ‘retrofitted’ to match the theme of the workshop.

**Results:**

Score of 4 (Strong): Submission is very well structured and follows all the criteria (i.e. clarity, novelty, interactivity, and coherency). However, practical significance/theoretical implications are not discussed.

**Reviewerconfidence:**

4. The reviewer's research focuses on human-centered ML.

**Reviewtext:**

## Summary

This paper proposes a framework to modularize output mathematics from deep neural networks based on the class inheritance paradigm in object-oriented programming. These mathematical formulation includes the variables, functions used in deep neural networks. The paper also includes a computer vision example as a use case for the proposed pipeline.

## Strengths

1. The proposed framework aims to improve accessibility of ML research, which is an often-overlooked challenge in ML community. I really appreciate authors’s effort in working on this problem.
2. The problem of output mathematics modularization is very interesting, and the proposed recommendations of using mixins to address this challenge is novel and promising.

## Minor weaknesses

1. The paper presents high-level guideline on how to modularize the output mathematics. Readers would find it more useful if there is an automatic/easy way to modularize output mathematics in their models, such as an open-sourced output mathematics modularization tool developed for PyTorch. It seems the authors are planning to release their code along with the camera-ready version (not available for reviewing).
2. The motivation for output mathematics modularization can be further improved, as the concept formulation of output mathematics can be unfamiliar for many readers. For example, the paper would be stronger if the introduction includes a concrete example where output mathematics can be beneficial.

**Score:**

Accept: The reviewer believes the submission provides a novel and reliable scheme to improve science communication but needs improvement.

---

### Official Review · AnonReviewer2 · 2021-03-30
**Accept with some revisions**

**Accessibility:**

Score of 3 (Neutral): Submission proposes methods to improve accessibility, but the level of intended accessibility is not well-articulated. Also, the limitations and exceptions are not stated.

**Litreview:**

Score of 2 (Needs Improvement): The submission leaves out prominent examples of previous work in the area.

**Problemstatement:**

Score of 4 (Strong): The submission sets a very strong example of how to address the problem, which should be relevant to the workshop themes.

**Relevance:**

Score of 4 (Strong): The submission directly addresses a theme of the workshop, and does so in a very professional manner.

**Results:**

Score of 3 (Neutral): Submission is well designed and provides a good level of coherency/novelty/interactivity.

**Reviewerconfidence:**

My confidence ranking is 4.

**Reviewtext:**

The authors address the challenge of "embedding the output mathematics (OM) in a modular component". They claim that their approach is the first practical solution to make the OM modular.

Their main contribution includes two recommendations to enable OMs to be modular which are:
1. Implementing OMs as methods associated with a set of base and mixin classes.
2. Reducing the constraints and interdependencies induced by the inclusion of arguments in the signature of methods, the variables that are relevant to the output mathematics should be part of the object instantiated from OMs class and be made accessible through its methods with as few parameters as possible.

In addition, to demonstrate the feasibility of the proposed method, they do a two step approach:
1. Applying their OM recommendations on the Panoptic-Deeplab implementation.
2. Applying modular OMs to prototype a variation of the instance segmentation formulation.

The paper topic is interesting and challenging. There are two concerns about this paper. Please check my following comments.
1. The paper needs to be proofread carefully.
There are several redundancies. For instance, the authors provide the same text for the main body and captions of figures. It's not common to write long captions in a technical writing. In addition, I propose to remove some sentences like "Reasons motivating such replacement are presented in the text" in the caption of Figure 5 and "See the text for the analysis and discussion of this table" in the caption of Table 1.

2. The authors refer to Appendix B that the overhead of their method is neglectable based on the peak memory overhead and computational overhead. There are comparisons based on the execution time and memory usage in the appendix. There is no a computational comparison!

Despite these problems, I recommend to accept the paper.



**Score:**

Accept: The reviewer believes the submission provides a novel and reliable scheme to improve science communication but needs improvement.

---

### Meta-Review · Area_Chair1 · 2021-04-01

**Recommendation:** Accept
**Confidence:** 4

**Metareview:**

This paper introduces a modularization scheme for grouping output mathematics (OM). The presented scheme results in greater flexibility to end users, such that the users may explore OM as freely as they would currently explore model specification code, etc.

Both the reviewers find the themes of the paper very relevant to the workshop. Reviewers also like the focus on accessibility and find this to be one of the first approaches towards modular OM. I agree with this broad reviewer assessment, and find that this addresses a much broader problem w.r.t. reproducibility. While the scope of conventional reproducible research efforts (e.g. the Neurips 2020 reproducibility check list) is to ensure the reproducibility of a single paper, this paper rightly notes that reproducibility efforts must also allow multiple independent papers to be evaluated on equal footing. This underlying principle makes me strongly recommend this paper be accepted to the workshop.

That said, I side with reviewer concerns on lack of computational evaluations. I also believe that the paper does not get to the primary talking points until too late into the manuscript. I would advocate for refurbishing the intro to a) provide a gentler exposition to OM, and b) clearly state the goals AND potential benefits (i.e., why should people care?).

---

### Decision · Program_Chairs · 2021-04-01

Accept (Poster)